# Green Synthesis of Blue-Emitting Graphene Oxide Quantum Dots for In Vitro CT26 and In Vivo Zebrafish Nano-Imaging as Diagnostic Probes

**DOI:** 10.3390/pharmaceutics15020632

**Published:** 2023-02-13

**Authors:** Govinda Gorle, Ganesh Gollavelli, Gowreeswari Nelli, Yong-Chien Ling

**Affiliations:** 1Department of Chemistry, National Tsing Hua University, Hsinchu 30013, Taiwan; 2Department of Humanities and Basic Sciences, Aditya Engineering College, Surampalem 533437, India; 3Jawaharlal Nehru Technological University, Kakinada 533003, India; 4Department of Chemistry, Andhra University, Visakhapatnam 530003, India

**Keywords:** graphene oxide, quantum dots, green synthesis, nano-imaging agents, diagnostic probes

## Abstract

Graphene oxide quantum dots (GOQDs) are prepared using black carbon as a feedstock and H_2_O_2_ as a green oxidizing agent in a straightforward and environmentally friendly manner. The process adopted microwave energy and only took two minutes. The GOQDs are 20 nm in size and have stable blue fluorescence at 440 nm. The chemical characteristics and QD morphology were confirmed by thorough analysis using scanning electron microscope (SEM), transmission electron microscope (TEM), atomic force microscope (AFM), Fourier transmission infra-red (FT-IR), and X-ray photoelectron spectroscopy (XPS). The biocompatibility test was used to evaluate the toxicity of GOQDs in CT26 cells in vitro and the IC_50_ was found to be 200 µg/mL with excellent survival rates. Additional in vivo toxicity assessment in the developing zebrafish (*Danio rerio*) embryo model found no observed abnormalities even at a high concentration of 400 μg/mL after 96 h post fertilization. The GOQDs luminescence was also tested both in vitro and in vivo. They showed excellent internal distribution in the cytoplasm, cell nucleus, and throughout the zebrafish body. As a result, the prepared GOQDs are expected to be simple and inexpensive materials for nano-imaging and diagnostic probes in nanomedicine.

## 1. Introduction

Recent research has focused on GOQDs because of their intriguing photoelectric and optical characteristics, which are brought about by their quantum confinement and edge effect [1,2]. The GOQDs have zero dimensions and range in size from 1–30 nm. Quantum confinement, a non-zero tunable band gap, a broad absorption spectrum between 270 and 370 nm, and size-dependent photoluminescent emission are some of the fundamental features of GOQDs [3]. The first GOQDs boom was initiated in 2010 by Pan et al. [4]. GOQDs have a higher crystallinity than their relatives and epitomize a unique class of carbon. GOQDs have excellent prospects for multiple applications including bioimaging, drug delivery, biosensors, environmental sensors, catalysis, energy conversion and harvesting [5,6,7,8,9].

Both the top-down approach and the bottom-up approach are used to prepare the GOQDs. Electrochemical redox, sonochemical destratification, chemical ablation, chemical exfoliation, and plasma treatment are all parts of the top-down approach [4,10,11,12,13]. Graphite [14], carbon nanofibers [15], carbon nanotubes (CNTs) [16], fullerenes [17], and other carbon sources [18] are frequently used as precursors to this strategy. According to the bottom-up approach, the carbon cores form from tiny molecules which typically are conjugated polycyclic aromatics or carbon atoms [19]. The size, shape, surface flaws (defects), and synthetic parameters of GOQDs all affect their chemical characteristics [20]. Both photoluminescence and UV-Vis absorption spectra are used to monitor the optical characteristics of GOQDs. The UV-Vis region is strong in all GOQDs’ UV-Vis spectra [21]. The physical attributes of GOQDs may be improved by tuning their usual band gap [22].

The hydrophilic properties, biocompatibility, chemical and photostability, and stable fluorescence (FL) emission inherent in GOQDs have led them to emerge as one of the most promising substitutes for highly toxic semiconductor QD in bioimaging, therapy, and sensing [22,23,24,25]. GOQDs are typically prepared by oxidizing, exfoliating, and then chopping large carbon precursors into nanosized pieces. Chemical oxidation, hydrothermal treatment, or solvothermal treatment has been used to prepare GOQDs. These processes involved extreme circumstances such as the continuous use of concentrated acids HNO_3_ or H_2_SO_4_ [1,4,26]. As a result, none of these synthetic processes was deemed “green” and all posed environmental problems. These processes also need to go through various processing stages to get acid residues out. The need for ecofriendly preparation of GOQDs is deemed to be indispensable.

Graphite [27,28], graphite oxide [29], graphene oxide (GO) [30], carbon and graphene nanofibers, and other specific carbon sources [31,32] are the most frequently used precursors in the synthesis of GOQDs. These materials are typically synthesized by following the Hummers method or its variations. However, the synthesis approach has certain drawbacks, including safety and environmental issues. It is very challenging to obtain a “metal-free” precursor despite multiple washings during the synthesis process. Hence, a substantial volume of concentrated H_2_SO_4_ or HNO_3_ is required. The accompanied production of hazardous gases such as N_2_O_4_ and NO_2_ during the oxidation process is inevitable [26]. The purity of the generated GOQDs may be impacted by the existence of trace metallic (Mn^2+^) contaminants from the KMnO_4_ precursor used for oxidation [33,34]. (2) This requires a long reaction and purification time. High concentrations of acid and inorganic salt residues have to be removed at the end of the reaction. Hence, finding alternative precursors is imperative.

Black carbon (BC), the world’s cheapest and most abundant wood material, is produced by heating *C. lanceolata* wood in minimal oxygen to remove water and volatiles. It has recently been investigated as a potential alternative precursor for the synthesis of GOQDs [35]. Yao et al. published a one-pot, eco-friendly hydrothermal technique for preparing GOQDs from BC [36]. Despite being effective in the preparation of GOQDs, it still needs high temperatures for a prolonged period (90 min). The synthesis of GOQDs from BC is being investigated here using a unique synthetic technique called microwave irradiation. Within 2 min, this technique generated GOQDs with diameters ranging from 20 to 25 nm. The synthesized GOQDs display strong photostability and great biocompatibility. The GOQDs have been effectively used as an acceptable FL probe. The GOQDs have been successfully demonstrated in FL nano-imaging of CT26 cells, as well as in zebrafish as an in vivo modal. This is the quickest and greenest approach for preparing GOQDs to date among the published works.

## 2. Materials and Methods

### 2.1. Materials and Reagents

The BC (Ningxia Yongruida Carbon Co., Ltd., Ningxia, China) was purchased from a local RT-MART. The reagents used are all analytical grade from Sigma-Aldrich (Merck Ltd., Taipei City, Taiwan) and were used straight away without any additional processing. Throughout the experimental work, 18.2 MΩ ultrapure water from the Merck Millipore pure water system (Merck Ltd., Taipei City, Taiwan) was employed.

### 2.2. Instrumental Measurements of GOQDs

A JASCO V-570 UV-Vis-NIR spectrometer was used to record the UV-Vis-NIR spectra in the 250 to 1200 nm wavelength range at 1 nm resolution. A JEOL JEM-2100 transmission electron microscope (HRTEM, 200 keV, JEOL, JEM-2100) and a scanning electron microscope (SEM, JEOL, JSM-7000F, FESEM) were used. High-resolution X-ray photoelectron spectroscopy (HRXPS) measurements were measured with a PHI Quantera SXM, ULVac-PHI Inc. A QM40 fluorescence spectrophotometer (PTI Ltd., Edmonton, AB, Canada) was used to record the fluorescence spectra. A Thermo Fisher Scientific K-Alpha 1063 X-ray photoelectron spectrometer (XPS, Thermo Fisher Scientific, Horsham, UK) was used for chemical analysis. A Perkin-Elmer system 2000 Fourier Transform Infrared spectrophotometer (FTIR) (Bomem model, DA-83FT) was used to record the FTIR spectra at 2 cm^−1^ resolution. An atomic force microscope (Agilent, 5100 PicoLE) was used to record the AFM images and measure the size and thickness of GOQDs.

### 2.3. Synthesis of GOQDs

The GOQDs were prepared using the microwave-assisted method with the following experimental conditions. A 100 mL thick glass vial was filled with 50 mg BC powder and 50 mL 1% H_2_O_2_, which was then placed into a household microwave oven (2.45 GHz) and irradiated at 600 W for 2 min at 30 s intervals (power off time gap). The chemical mixture obtained after the reaction was cooled to room temperature and centrifuged at 8000 rpm. The synthesized GOQDs were collected from the supernatant.

### 2.4. Determination of the Fluorescence Quantum Yield of GOQDs

The fluorescence quantum yield (Φ_x_) of the synthesized GOQDs was calculated [37] using the following equation:Φ_x_ = Φ_r_(F_X_/F_r_)(A_r_/A_x_)(n_x_/n_r_)^2^

where Φ, F, A, and n respectively represent the FL quantum yield, integrated FL intensity of the emitted light, the absorbance of the solution, and the refractive index of the solvent. The subscripts “x” and “r” refer to the unknown and reference fluorophore, respectively. Herein, we employed quinine sulfate (whose Φ_x_ is 0.54 when dissolved in 0.1 M H_2_SO_4_). The absorbance of GOQDs and quinine sulfate solution in a 10 mm fluorescence cuvette were adjusted to never exceed 0.05 at the excitation wavelength to reduce the effects of re-absorption. The quinine sulfate was dissolved in 0.1 M H_2_SO_4_.

### 2.5. Biocompatibility Assays

Using the MTT (3-(4, 5-dimethylthiazol-2-yl)-2,5-diphenyltetrazolium bromide) assay, the biocompatibility of GOQDs on CT26 cells (the murine colorectal carcinoma cell line, CT26 WT, ATCC^®^ CRL-2638TM) obtained from ATCC (American type culture collection) was used for assessment [38]. Each well containing 2.0 × 10^4^ cells in 1 mL of RPMI-1640 medium was taken into a 24-well plate and cultured for 24 h. The 24-well plate was then filled with various concentrations (0 to 200 μg/mL) of blue-emitting GOQDs aqueous solution followed by culturing for a further two days. The 24-well plate was filled with 50 µL of MTT reagent (0.5 mg/mL). The incubation was extended for another 4 h after the cells had been treated with GOQDs for 2 days. The top layer of the solution in each plating cell was then removed and a 1 mL DMSO aliquot was added to each well thereafter. Appropriate pipette stirring was carried out to dissolve the violet formazan product. The finished solution was then centrifuged in each well for five min at 13,000 rpm to remove any remaining solids. Finally, the optical absorbance at 570 nm was measured. The optical absorbance of the control and the GOQDs in comparison with the standard curve was converted to cell viability.

### 2.6. In Vitro Imaging of GOQDs by Confocal Laser Scanning Microscope (CLSM)

The seeding of the cells for pretreating the CT26 cells with GOQDs was the same as previously discussed in Section 2.5, and the further steps were as follows. Briefly, PBS was used to clean the blue-emitting GOQDs internalized in CT26 cells. The cells were rinsed three times with PBST (5% Tween 20 in PBS) and fixed on a glass slide with paraformaldehyde solution (4%) in PBS for 5 min. Triton X-100 solution (1 mL) was then added. The cells were stained with DAPI (4′,6-diamidino-2-phenylindole) (1 ng/mL PBS, 30 min) after being exposed to Triton X-100 for 1 h. The cells were imaged using a CLSM (Zeiss LSM 700, Oberkochen, Germany) equipped with a UV laser (e-Max 405 nm; detector 420–450 nm) and an Ar laser (e-Max 488 nm; detector 510–540 nm).

### 2.7. In Vivo Zebrafish Compatibility and Fluorescence Imaging of GOQDs

All in vivo tests were conducted using male and female zebrafish (*Danio rerio*) of the wild type and AB strains [39]. The in vivo biocompatibility and nano-imaging of GOQDs were assessed on wild-type and AB strains of male and female zebrafish. Their use was reviewed by the Taiwan Zebrafish Core Facility and approved by the Ethical Committee of Affidavit of Approval of Animal Use Protocol (approval on 20 March 2014) by the National Tsing Hua University. All animal experiments were performed in compliance with the Animal Management Rules at National Tsing Hua University. The zebrafish were kept in a fish container at 28 °C with regulated light and dark cycles of 10 h and 13 h, respectively. Fresh zebrafish seeds (embryos) were harvested for the tests on biocompatibility and fluorescence imaging. Zebrafish seeds were distributed into groups of ten and given separate treatments with various concentrations of GOQDs (0, 25, 50, 100, 200, and 400 μg/mL). Based on the counting of the survived embryos at various periods of development from 3 to 96 hpf (hour-post fertilization), the in vivo toxicity was determined. The embryonic development from 3–96 hpf was observed using a bright field microscope (inverted model) with a DP73 digital camera (SZX10, Olympus, Tokyo, Japan). Using a fluorescent microscope, the zebrafish fluorescence images were captured (Olympus IX71, Tokyo, Japan). The following formulas were used to predict the survival rates at a hatching time of 96 hpf.
Embryo survival rate (%) = Viable embryos no/total embryos no × 100%
Hatching rate (%) = Hatched embryos no/embryos total no × 100% 

## 3. Results and Discussion

### 3.1. Synthesis and Characterization of GOQDs

BC and hydrogen peroxide were used to prepare the GOQDs in a single pot with microwave assistance. The following benefits of the oxidant H_2_O_2_ are anticipated: (1) When exposed to microwave radiation, H_2_O_2_ is broken down into free radicals such as •OH and •O. Due to their powerful oxidizing abilities and high reactivity, these free radicals may easily oxidize and cleave the graphene structure of BC, negating the need for a strong concentrated acid to achieve robust oxidation; (2) It prevents the release of harmful gases which are toxic to the environment; and (3) H_2_O_2_ has only H and O as its constituent elements, preventing contamination from even minute amounts of metallic impurities that occurs in the case of using KMnO_4_. The advantage of the microwave technique was that it is green, required less skill for operation, and took a very short time to react (just 2 min). The same microwave technique has been adopted by Romero et al. recently by considering citric acid as a starting material to produce carbon dots for the in vitro and in vivo phototherapy of bacteria. However, the process takes 8 min to complete [40]. By shortening the time, Kang et al. produced GOQDs from coal in 5 min for bioimaging. In comparison with other microwave methods, the current report is advanced in terms of process time and the green chemicals’ adaptation for the preparation of graphene QDs (GQDs) [41]. Table 1 shows the comparison between various preparation methods of GOQDs along with the current report [42,43,44,45,46,47,48,49,50,51,52,53,54,55].

SEM, TEM, and AFM were used to analyze the as-prepared GOQDs’ morphology. Before microwave treatment, BC appeared as a wrinkled and folded sheet-like structure of around 5 μm as seen in the SEM and TEM images in Figure 1a,b, respectively.

However, following microwave irradiation, GOQD diameters substantially shrank to ~20 nm with extremely thin morphology, as shown in Figure 2a–c. The nano size and thin dot/sphere-like morphology have been demonstrated with low and high magnification images of TEM (Figure 2a,b). The ring-like electron diffraction patterns from Figure 2c represent the poly crystallinity of the GOQDs. The GOQDs’ AFM images in Figure 2d,e reveal the rough topography with dot-like structural features and a size of ~20 nm. This is comparable to the HRTEM results. The height distribution of ~3–6 nm along with some aggregates are seen in Figure 2e (a 3D reconstructed image). The height represents the normal thickness of GO (3 nm) and a few layered GO (6 nm) and indicates that the GOQDs are made of thin graphene GO layers [56].

After morphological and physical structural conformation from electron microscope techniques, FT-IR spectral recordings were used to identify the GOQDs’ chemical functionality. As seen in Figure 3a, the strong band for aromatic C=C stretching vibration was seen at 1590 cm^−1^, and two other bands for the aromatic C-H out-of-plane deformations were seen between 873 cm^−1^ and 753 cm^−1^. Additionally, the existence of the C-O, C-O-C, C-OH, C=O, and -OH groups was indicated by the peaks at approximately 1087 cm^−1^, 1255 cm^−1^, 1381 cm^−1^, 1719 cm^−1^, and 3411 cm^−1^ [57]. These oxygen-containing functional groups are an indication of the formation of GOQDs (successful oxidation of BC) and their hydrophilic properties help to keep GOQDs stable in aqueous solutions. Additionally, a faint C=O peak at 1719 cm^−1^ in the FTIR spectrum of BC revealed that BC had fewer oxygen functional groups. The FTIR spectrum of GOQDs, however, showed that this peak (1719 cm^−1^) was more pronounced, indicating that GOQDs possessed more oxygen-containing groups than BC. The XPS spectrum provides additional confirmation of the GOQDs’ chemical composition. Figure 3b (survey scan spectrum GOQDs) revealed clear peaks at about 284.9 eV and 532.4 eV corresponding to the C1s and O1s, respectively. The binding values of C=C at 284 eV, C-OH at 286.0 eV, C=O at 287.8 eV, and O-C=O at 289 eV are significant for carbon, as it contains oxygen functional groups from GOQDs that are demonstrated in the C1s deconvolution spectrum (Figure 3c) [46]. A high-intensity signal at 533.3 eV in the HRXPS O1s spectrum of GOQDs (Figure 3d) further supports the richness of the oxygen-related functional groups and the profound oxidation during microwave irradiation. The deconvolution results reveal that the single broad O1s peak further displays two distinct peaks at 531.6 eV and 533.1 eV, corresponding to the C=O and C-OH oxygen functional groups on the surface of the GOQDs derived from BC [58,59]. The peak at 531.6 eV has less intensity than at 533.1 eV, which is an indication that the GO surface was less enriched with carbonyl functional groups than alcohol and epoxide types of oxygen functionalities. The computed C/O ratio of 1.84 indicates that GOQDs have undergone additional oxidation, which is consistent with the FT-IR findings. The TEM, AFM, XPS, and FT-IR results are in good agreement with previous reports demonstrating the morphology, size, and surface chemistry of the GOQDs prepared by microwave, hydrothermal, and electrochemical methods [60].

### 3.2. Optical Properties of GOQDs

Figure 4a depicts the UV-Vis absorption of GOQDs solution showing that GOQDs displayed two characteristic absorption peaks related to the GO structure. The peak at ~220 nm corresponding to π→π* transition is the indication of C=C and the peak at roughly 320 nm is related to n→π* transitions of C=O in GOQDs. The UV-Vis absorption consists of excitation and emission peaks at 300 nm and 440 nm. The blue PL emission characteristic of GOQDs was then examined. The synthesized GOQDs displayed maximum emission at 440 nm when excited with a wavelength of 320 nm, as shown in Figure 4b. The FL quantum yield was determined by using the reference compound quinine sulfate (in 0.1 M H_2_SO_4_, the quantum yield was 54%) and computed for GOQDs at about 3.13%. As shown in Figure 4b the GOQDs FL was excitation-dependent. It was suggested that the defects and functional groups present on the surface may be responsible for the FL emission [61].

GQDs and GOQDs have exhibited superior PL properties for bioimaging as conventional inorganic and organic fluorophores. They have inspired researchers to use them for cancer cell bioimaging. Table 2 lists various synthesized GQDs and GOQDs as well as their physical parameters such as size, quantum yields, and color of luminescence [62,63,64,65,66,67,68,69,70,71]. The reported GOQDs synthesized by the green synthesis method have blue emission at 440 nm with 20 nm size, which is suitable for in vitro and in vivo imaging.

### 3.3. In Vitro Biocompatibility and Nano-Imaging of GOQDs in CT26 Cells

The cytotoxicity of the GOQDs was assessed on CT26 cells using the MTT assay at different dosages (0 to 200 μg/mL). The toxicity profiles are very important before examining the potential use of GOQDs for nano-imaging applications. The results revealed that the cell viability remained over 85% for all concentrations tested (Figure 5a) signifying good in vitro biocompatibility of the GOQDs. Kang et al. and Cunci et al. have reported that most of the GOQDs are significantly less toxic to mammalian cells and have the highest biocompatibility [46,72]. Hence, the GOQDs prepared in this work are potential candidates for use in nano-imaging applications. The GOQDs are later added to the CT26 cells for 24 h before being examined with the CLSM. After being incubated with GOQDs, the CT26 cells displayed blue fluorescence, as shown in Figure 5c,d. These images show that the synthesized GOQDs are internalized through a non-targeted endocytosis internalization mechanism and are well distributed throughout the cytoplasm and nucleolus [73].

The advantage of GOQDs as a nano-imaging agent is the smaller size, which can facilitate the cell membrane crossover, as well as their very good water solubility in the biological media with less toxicity than other carbon materials and metal and metal oxide nanomaterials [74]. Apart from this, the surface π bonds and oxygen functional groups at the edges can help to functionalize different biomarkers and other potent organic drugs through covalent and non-covalent approaches for efficient theranostics [75]. Previously, we have prepared multifunctional magnetic graphene by functionalizing the GQDs with Gd, folate, and doxorubicin (folate–GdGQD/Dox) for targeted imaging and the therapy of HeLa and HepG2 cells. The material has shown excellent MRI, CLSM imaging, and good DOX loading, thereby promising therapeutic capability [54]. In addition, we have also explored the microwave synthesis of multifunctional MGQDs from crab shells with different metals such as Gd^3+^, Mn^2+^, and Eu^3+^ for MRI and CLSM imaging followed by targeted cancer therapy. The MGQDs have shown excellent toxicity against cancer cells compared to normal cells [55]. Herein, we have developed simple, green, and metal-free GOQDs for nano-imaging in vitro and in vivo, which is very affordable compared to the previous works reported in Table 1 and Table 2. We are anticipating that this method could be one of the best to synthesize the GOQDs-based nanodrugs for future nano-imaging and therapeutic applications.

### 3.4. In Vivo Biocompatibility and Nano-Imaging of GOQDs in Zebrafish Embryos

The in vivo biocompatibility and nano-imaging of GOQDs were assessed on wild-type and AB strains of male and female zebrafish (*Danio rerio*). Their use was reviewed and permitted by the Taiwan Zebrafish Core Facility at National Tsing Hua University. The zebrafish were raised in an aquarium at 28 °C under controlled 10 h/13 h light/dark conditions. All animal experiments were performed in compliance with the Animal Management Rules of the National Tsing Hua University.

*Danio rerio* is a precious non-mammalian vertebrate model that is frequently used to investigate diseases such as cancer. The significant evolutionary preservation of cancer-related genes among zebrafish and humans enables the extrapolation of study findings from fish to people. These species have been a reliable model for cancer research due to their high fecundity, economical maintenance, feasibility to dynamically visualize tumor progression in vivo, and for the toxicity evaluation of drugs at affordable prices. To understand the pathophysiology of a complicated disease like cancer, it may be crucial to look at both genetic and epigenetic changes. Zebrafish have proven to be a trustworthy animal system to use for research on human cancer. Rapid, extensive analyses of zebrafish in vivo drug reactions and kinetics will likely result in novel applications in combination therapy and personalized medicine. Zebrafish are on the verge of joining the mouse as a pre-clinical cancer model due to all of the aforementioned factors [76].

For our preliminary in vivo imaging studies, zebrafish embryos were used as a live model at various developmental stages, including the cleavage (3 hpf), pharyngula (24 hpf), hatching (48 hpf), and larval (96 hpf) stages. The images of zebrafish embryos exposed or not (0 mg/L) to GOQDs are displayed in Figure 6. The findings showed that even at high concentrations, GOQDs had no harmful effects on embryonic development and had no effect at any stage of development at all. In addition, the developed fish showed the same morphology despite having different levels of GOQDs exposure and control experiments. The reported values agreed with our previously reported zebrafish biocompatibility studies with MG/MFG and other carbon nanomaterials [77,78,79]. The imaging and bio-compatibility experiments led by Kang et al. have also reported that there is no effect of the carbon quantum dots (CQDs) on the growth of the zebrafish and showed good distribution throughout the zebrafish body [79]. Our preliminary in vivo biocompatibility results suggest the potential of GOQDs for future nano-imaging and diagnostic applications as a theranostic delivery system for carrying various drugs [54].

Zebrafish embryos were employed as a model for the in vivo imaging application in order to confirm the presence and distribution of GOQDs. A well grown zebrafish embryo is shown in Figure 7a, with its entire body without GOQDs solution at 96 h as a control. Zebrafish embryos treated with 100 μg/mL GOQDs show an improvement in color which demonstrates that the GOQDs were successfully internalized and sustained to the larvae (Figure 7b). GOQDs might enter the embryo’s body via skin absorption or swallowing. In Figure 7b, the fluorescence image of zebrafish shows that the dorsal aorta is bright, indicating that GOQDs have entered the circulatory system, which is crucial for GOQD transport in zebrafish. Skin absorption is a significant way for GOQDs to enter zebrafish, and their metabolism might eliminate some GOQDs from the zebrafish embryos. Therefore, the fluorescence released by GOQDs serves as an illustration of the zebrafish’s outline. Thus, the distribution of GOQDs in zebrafish reveals the ADME (absorption, distribution, metabolism, and excretion) route. The brightness of the blood vessels and the tissue in the tail provides evidence that certain GOQDs enter the cardiovascular system and spread throughout the body. In this sense, GOQDs are a type of in vivo imaging-appropriate biocompatible probe without obvious quenching. Zebrafish and mammals share many biological similarities, hence the findings from zebrafish are utilized to simulate biological consequences in other higher animals. Xu et al. prepared N-doped CDs from urea, aniline, and ethylenediamine as a starting material. The prepared N-doped CDs have exhibited green fluorescence and demonstrated Hg^2+^ FL sensing and imaging in the zebrafish [78]. Another study by Yue et al. prepared red luminescent CQDs by Ru doping. The resultant Ru-CQDs were also demonstrated for zebrafish imaging and the photodynamic therapy of cancer [80]. In comparison, the GOQDs prepared in this work have very good luminescence, which is similar to or better than these green and red CQDs reported. CQDs have a pronounced importance in curing brain cancers by minimizing the limitations of the blood-brain barrier (BBB). The BBB completely blocks nearly all drugs targeting brain neoplasms, which has been one of the major challenges for brain cancer treatments. The tunable and small sizes of CQDs with excellent luminescence could be of great advantage in treating the brain cancers reported [81]. In addition to CQDs ultrasmall size, shape, solubility, and biocompatibility, they can be triggered by external stimuli such as pH, temperature, ultrasound, and near-infrared (NIR) light for the controlled release of drugs, all of which are added advantages in nanomedicine and future theranostics applications [82]. Moreover, the CQDs have excellent biodistribution with great biocompatibility compared with other nanomaterials [83]. Hence, we are anticipating that the GOQDs presented in the study will provide an important contribution in cancer diagnosis and therapy. We would like to investigate the elimination of colon cancer in zebrafish using modified GOQDs, which could serve as an imaging and therapeutic probe in our future studies.

## 4. Conclusions

In summary, we used BC as the starting material and green oxidant H_2_O_2_ in a one-pot microwave-assisted oxidation process to successfully prepare GOQDs. Under microwave circumstances, H_2_O_2_ produced free radicals that can also break huge BC to form GO fragments and GOQDs. The benefits of this synthetic approach are as follows: (1) the method can avoid using strong concentrated acid and producing poisonous fumes; (2) the strategy can prevent metal impurity contamination; (3) it only involves affordable, environmentally friendly reagents; and (4) the synthesis process is very simple and only takes a short time (2 min). The formation of GOQDs is confirmed by electron microscopic and spectroscopic techniques. The synthesized GOQDs shown good photo-stability and admirable biocompatibility in vitro against CT26 cancer cells and in vivo zebrafish (*Danio rerio*) embryo model. The FL imaging of CT26 cancer cells and zebrafish indicate that the GOQDs can be excellent nano-imaging and diagnostic probes with biocompatibility.

## Figures and Tables

**Figure 1 pharmaceutics-15-00632-f001:**
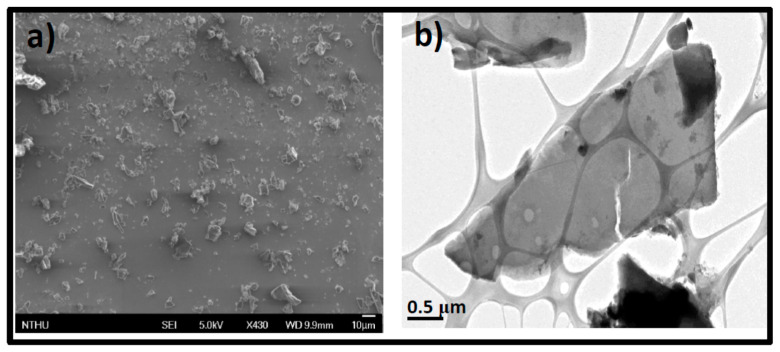
Electron microscope images of BC before microwave irradiation (**a**) SEM. (**b**) TEM.

**Figure 2 pharmaceutics-15-00632-f002:**
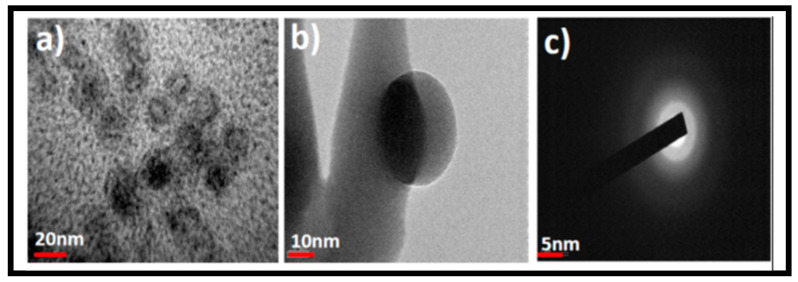
HRTEM images of synthesized GOQDs (**a**) Low magnification (scale bar, 20 nm). (**b**) High magnification (scale bar, 10 nm). (**c**) The electron diffraction pattern recording. (**d**) AFM of synthesized GOQDs and its 3-D imaging profile. (**e**). The AFM was recorded at the length of 500 nm.

**Figure 3 pharmaceutics-15-00632-f003:**
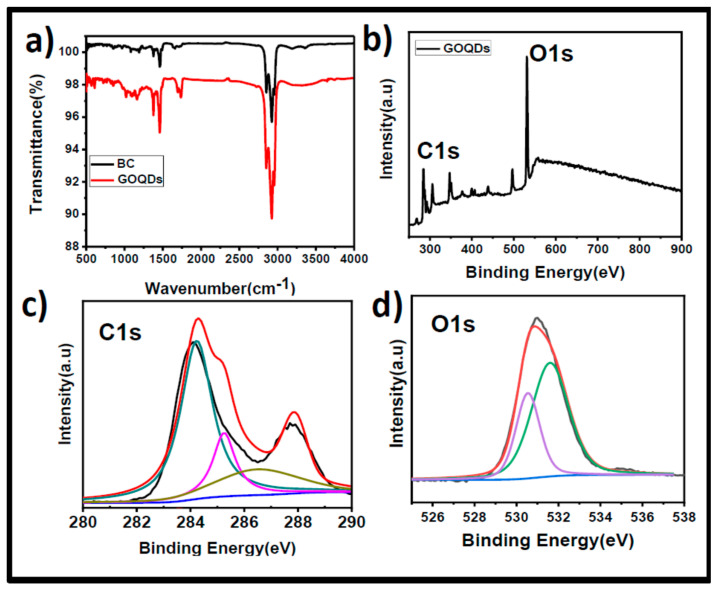
(**a**) FTIR spectra of synthesized GOQDs. (**b**) XPS survey spectrum of GOQDs. (**c**) C1s deconvolution XPS spectra of GOQDs. (**d**) O1s deconvolution XPS spectra of GOQDs. The black color line is for the original spectrum and the other colored lines are the deconvoluted spectral lines in both subfigures c and d.

**Figure 4 pharmaceutics-15-00632-f004:**
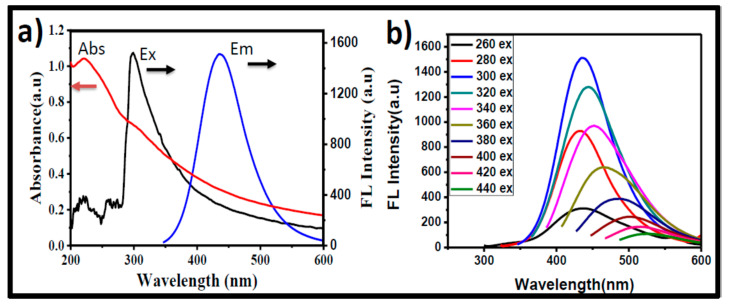
(**a**) UV−Vis absorption, fluorescence excitation and emission spectra of the green synthesized GOQDs. The arrows indicate the direction towards the respective ax (absorption or FL). (**b**) Corresponding fluorescence spectra of GOQDs with variable intensities at different excitation wavelengths ranging from 260 nm to 440 nm.

**Figure 5 pharmaceutics-15-00632-f005:**
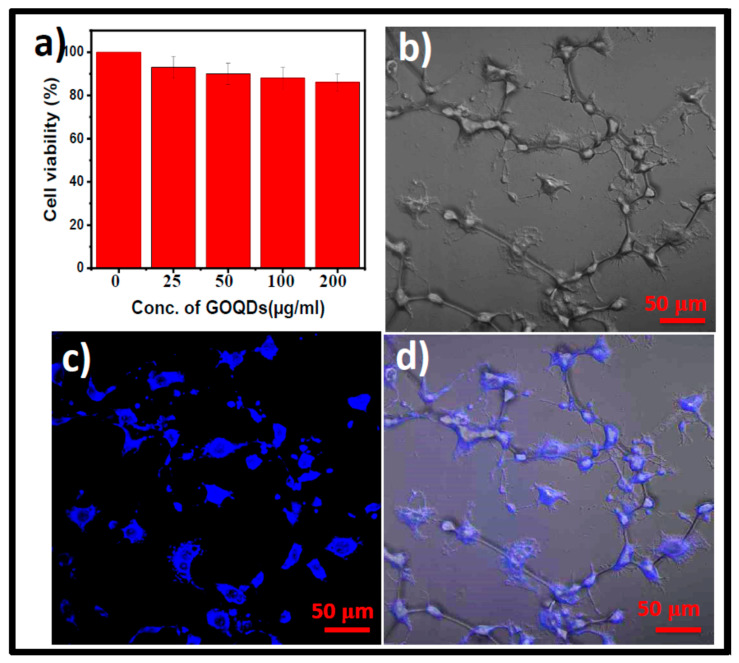
(**a**) Cell viability assay of GOQDs at different concentrations. Confocal fluorescence images of GOQDs in CT26 cells. (**b**) Bright-field image. (**c**) Blue emission of GOQDs within the cells recorded at 405 nm excitation. (**d**) The overlay of bright-field (**b**) and fluorescence (**c**) images. (The concentration of GOQDs used was 200 µg/mL).

**Figure 6 pharmaceutics-15-00632-f006:**
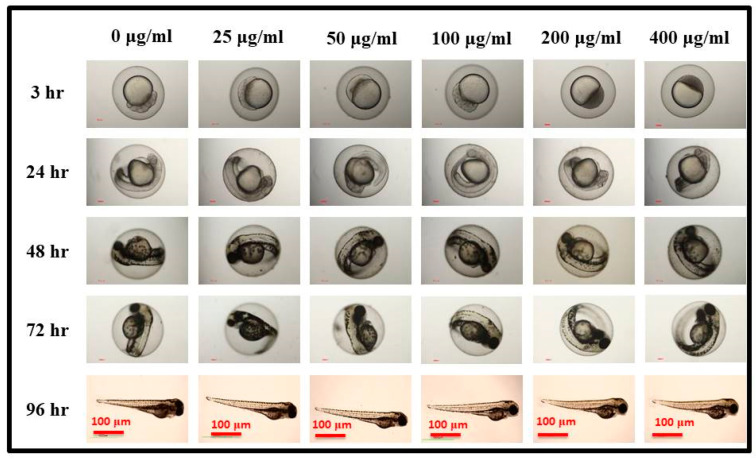
Optical microscopy images of representative developmental stages of zebrafish embryos in water containing different concentrations of GOQDs at different times. Scale bar 100 µm.

**Figure 7 pharmaceutics-15-00632-f007:**
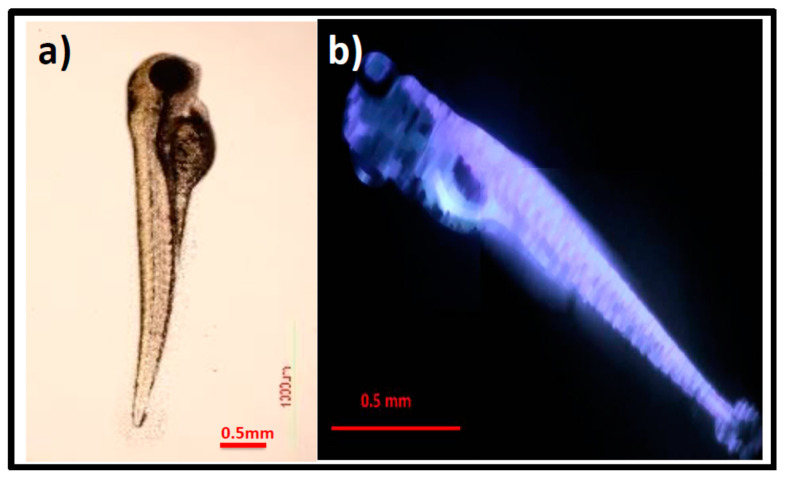
(**a**) Control (0 μg/mL) optical image of zebrafish. (**b**) Fluorescence image of GOQDs treated zebrafish (100 μg/mL). The images recorded at 96 hpf.

**Table 1 pharmaceutics-15-00632-t001:** Methods for synthesizing GOQDs.

S. No	Precursor	Synthetic Process (Method, Reagents, Time)	Post-Processing Steps	Time	Ref. No
1	Graphite Powder	Step 1: Oxidation method H_2_SO_4_/KMnO_4_/NaNO_3_/H_2_O_2_, >1 d Step 2: Hydrothermal method, Ndimethylformamide, 6 h	Dialysis, filtration, and rotary evaporation	>8 h	[42]
2	Graphite	Step 1: Oxidation method H_2_SO_4_/KMnO_4_/NaNO_3_/H_2_O_2_, >1 d Step 2: Acid refluxing, HNO_3_, 8 h Step 3: Ultrasonic, NaOH, 0.5 h	Centrifugation, dialysis, and filtration.	>1 week	[43]
3	Graphite Nanoparticles	Step 1: Ultrasonication H_2_SO_4_/HNO_3_, Step 2: Acid refluxing, H_2_SO_4_/HNO_3_ 12 h	Centrifugation, and dialysis	>2 d	[44]
4	Graphene nanofibers	Step 1: Acid oxidizing, NaClO_3_, fuming HNO_3_, 12 h Step 2: Acid refluxing, NaClO_3_, fuming HNO_3_, HNO_3_, 6 h Step 3: Ultrasonication, HCl, 1 h	Dialysis and vacuum drying	1 week	[45]
5	Coal	The coal and ethanol added in 50 mL of Bottom of a glass vessel, and applied Q-switch ND:YAG laser system, pulsed laser beam with an ablation energy of 0.1 J.	Filtered microporous membrane and dialysis for two days	5 min	[46]
6	Graphite	Step 1: Oxidation by Hummers method, H_2_SO_4_/KMnO_4_/NaNO_3_/H_2_O_2_, >1 d Step 2: further oxidation by Hummers method, H_2_SO_4_/KMnO_4_/NaNO_3_/H_2_O_2_, >1 d	Centrifugation, and dialysis.	>2 d	[47]
7	XC-72 carbon black	Step 1: Ultrasonication, HNO_3_, 2 h Step 2: Acid refluxing, HNO_3_, 24 h	Centrifugation, evaporation, and redispersion	>24 h	[48]
8	XC-72 carbon black	Acid refluxing, HNO_3_, 24 h	Centrifugation, vacuum filtration, and vacuum freeze drying	>24 h	[49]
9	Coal	Acid refluxing, HNO_3_, 12 h	Centrifugation, vacuum dry, and redispersion	>12 h	[50]
10	Starch	Starch solution in water, Hydrothermal method at 190 °C for 120 min.	Centrifugation, and dialysis	2 h	[51]
11	Durian extract	Durian flesh suspension in water, Hydrothermal method at 150 °C for 12 h.	Filter membrane, and centrifugation.	12 h	[52]
12	Lignin	Lignin acidic solution, Hydrothermal method at 180 °C for 12 h.	Filter membrane, and centrifugation	12 h	[53]
13	Crab shells	Step 1: Dried crab shells powder in acetic acid solution by stirring for 12 h at TR Step 2: The upper layer solution and GdCl_3_ added in glass vessel then heated using a single-mode microwave reactor at 220 °C for 10 min.	Filter membrane, and centrifugation	12 h	[54]
14	Graphite	Step 1: oxidation by Hummer’s method, H_2_SO_4_/KMnO_4_/NaNO_3_/H_2_O_2_, >1 d Step 2: Graphene oxide in water was hydrothermally treated at 120 °C for 10 h. GQDs were purified by membrane filtration.	Filter membrane, and centrifugation	22 h	[55]
15	Black carbon	Microwave-assisted method. No chance of heavy metal contamination and required excessive purification with acids or other solvents	Simple membrane filtration.	2 min	This work

**Table 2 pharmaceutics-15-00632-t002:** Physical parameters of synthesized GQDs/GOQDs.

S. No	Materials	Size	PL Color	Emission (nm)	Quantum Yield (%)	Ref.
1	GQDs	2−6	Red	600	1.4	[62]
2	GQDs	3−5	Blue	427, 516 (shoulder)	4.1	[63]
3	GQDs-Cyc-HCl	5	Green	400−500	58	[64]
4	N-GQD/N−S-GQD	5.5−3.9 avg.	Green	425, 448	60	[65]
5	Carbon dots (CDs).	3	Blue	442	25	[66]
6	GQD-RGD	4	Blue-green	420	NA	[67]
7	GQD-amine	7.5	Green	500	1.5	[68]
8	GQD-RGD	3.7	Yellow-green	460	NA	[69]
9	Graphene quantum dots (GQDs)	5	Blue	430	23.8	[70]
10	GQDs	6	Blue	400	28	[71]
11	GOQDS	20	Blue	400	3.13	Present work

## Data Availability

Data is contained within the article.

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
