# Peer review of "Green Synthesis of Blue-Emitting Graphene Oxide Quantum Dots for In Vitro CT26 and In Vivo Zebrafish Nano-Imaging as Diagnostic Probes"

_pharmaceutics, 2023, doi:10.3390/pharmaceutics15020632_

Round 1

Reviewer 1 Report

This work has synthesized the blue-emitting GOQDs and applied them the in vitro and in vivo imaging. The experiments were with poor logic and incomplete.

1. The word usage is not precise, for example, “Quantum dots are the smallest nanoparticle…”, despite their variable sizes, iron oxide also has a small size below 10 nm.

2. The citations in the first paragraph of the introduction have too many reviews, the history of GOQDs should be told by citing the original references.

3. The fourth paragraph of the introduction is redundant.

4. What is the source of the black carbon used in this work? How did the authors get it (self-made or purchased)?

5. What is the diameter of black carbon?

6. The power of the microwave oven is 600 W, how did the authors operate the preparation of GOQDs with a power of 800 W? What is the meaning of “30 s intervals”? What is the centrifuge parameter used in the preparation of GOQDs? Is there any additional treatment after the extraction of the supernatant?

7. What are the shadows in Figure 2 (a)?

8. The size distribution and height distribution of GOQDs should be given.

9. What is the meaning of “GO” in XPS characterization?

10. The HRXPS of O1s spectra of GOQDs should be analyzed.

11. Why do the authors conclude that “The reported GOQDs synthesized by the green synthesis method has excellent blue emission at 400 nm with 25 nm size, which is greatly suitable for many biological applications such as in vitro and in vivo imaging and therapy”?

12. Are the dosages of GOQDs chosen for cytotoxicity experiments suitable? What is the basis for the dosage choice? In a real application, what dosage of GOQDs will take to achieve the best performance?

13. Will the zebrafish emit fluorescence under UV radiation? Will the background fluorescence disturb the experiments?

14. What is the relation between CT-26 and zebrafish?

Author Response

Point-by-point response to Comments and Suggestions from Reviewer 1

Thank you for reviewing our work and positive comments. We have made the suggested changes in the revised manuscript which are highly important to improve the quality and readability of the manuscript.

  1. The word usage is not precise, for example, “Quantum dots are the smallest nanoparticle…”, despite their variable sizes, iron oxide also has a small size below 10 nm.

Response:

Thank you for your constructive comment. QDs are very unique in size and color. So, we have given a general introduction before stating our discovery. The authors do not mean that QDs are only having a smaller size. We have removed the text from the manuscript to avoid confusion first. We further rewrite the abstract (line no. 11-23) and the title (line no. 2-3) to present the highlight of our work. Please check line no 13 for revision. The same has been here for your reference.

Green synthesis of graphene oxide quantum dots for in vitro and in vivo biocompatible fluorescent imaging probes

Abstract: Graphene oxide quantum dots (GOQDs) are prepared using black carbon as a feedstock and H2O2 as a green oxidizing agent in a straightforward and environmentally friendly manner. The process adopted microwave energy and just takes two minutes. The GOQDs are 20 nm in size and have stable blue fluorescence at 440 nm. The chemical characteristics and QD morphology were confirmed by thorough analysis using SEM, TEM, AFM, FTIR, and XPS. The biocompatibility test was used to evaluate the toxicity of GOQDs in CT26 cells in vitro and the IC50 was found to be 200 µg/mL with excellent survival rates. Additional in vivo toxicity assessment in the developing zebrafish (Danio rerio) embryo model found no observed abnormalities even at a high concentration of 400 μg/mL after 96 hours post fertilization. The GOQDs luminescence was also tested both in vitro and in vivo. They showed excellent internal distribution in the cytoplasm, cell nucleolus, and throughout the zebrafish body. As a result, the prepared GOQDs are expected to be simple and inexpensive materials for nano-imaging and diagnostic probes in nanomedicine.

  1. The citations in the first paragraph of the introduction have too many reviews, the history of GOQDs should be told by citing the original references.

Response:

Thank you for your valuable suggestion. Our intension of citing reviews is to give a broad view and understanding about the material. Per your suggestion, we have deleted 7 review references and replaced them with original references. Please check lines no 398 to 431 for revised references. The same has been here for your reference.

  1. The fourth paragraph of the introduction is redundant.

Response:

Thank you for your constructive comment. The manuscript is based on the synthesis of the green GOQDs from BC. Hence, it is very important to highlight the existing GOQDs sources and the drawbacks in the existing literature. This paragraph is one of the key pieces of information for the readers. They will be aware of the existing GOQDs raw materials, chemicals, and methods adopted for preparation and existing advantages and disadvantages if any. Lastly, we conclude the paragraph with “High concentrations of acid and inorganic salt residues have to be removed at the end of the reaction. Hence, finding alternative precursors is imperative.” Hence, we request the reviewer understand the importance of the paragraph. If there are any key objections to the content, we are glad to avoid them in the final version.

  1. What is the source of the black carbon used in this work? How did the authors get it (self-made or purchased)?

Response:

Thank you for your informative question. The BC (Ningxia Yongruida Carbon Co., Ltd. China) was purchased from local RT-MART. It was not self-made. Please check line no 87. The same has been here for your reference.

  1. What is the diameter of black carbon?

Response:

Thank you for your informative question. The diameter of black carbon is around 5 μm after grounding to small pieces (From Figures 1a and 1b). Please check line no 192. The same has been here for your reference.

  1. The power of the microwave oven is 600 W. How did the authors operate the preparation of GOQDs with a power of 800 W? What is the meaning of “30 s intervals”? What is the centrifuge parameter used in the preparation of GOQDs? Is there any additional treatment after the extraction of the supernatant?

Response:

Thank you for your critical question. We have corrected the typo. The microwave power used in the experiment is 600 W. While irradiating the sample inside the microwave oven, we have given some gap (power-off) every 30 seconds until we get the GOQDs. After the successful preparation of the GOQDs from microwave irradiation, we centrifuged the solution at 8000 rpm to remove the non-oxidized BC deposits and the supernatant contains the GOQDs. We did not process the GOQDs further and used them directly without any modifications to the toxicity and imaging studies. Please check line no 109. The same has been here for your reference.

  1. What are the shadows in Figure 2(a)?

Response:

Thank you for your informative question. The sphere-like shadows are the GOQDs obtained in the experiments. We replace the previous image with a new one for your reference. The shadows in Figure 2a are no of GOQDs aggregate on a TEM Cu grid. Please check line no 209. The same has been here for your reference.

  1. The size distribution and height distribution of GOQDs should be given.

Response:

Thank you for your valuable suggestion. The size distribution and height profiles are taken from TEM and AFM and mentioned in the revised manuscript. Please check lines no 197 to 206. The same has been here for your reference.

  1. What is the meaning of “GO” in XPS characterization?

Response:

Thank you for your informative question. The GO mentioned in XPS is refers to Graphene Oxide. It has to mention as a GOQDs. However, excluding the size and luminescence, the carbon chemical functionality resembles GO. As a result, we denoted the GO with GOQDs instead. Please check line no 223.

  1. The HRXPS of O1s spectra of GOQDs should be analyzed.
    Response:

Thank you for this valuable suggestion. We observed the two major peaks in the survey scan spectra of GOQDs. One at 284 eV for C1s and another peak at 533.3 eV for O1s. The broadening of these peaks are associated with many other peaks corresponding to various oxygen functional groups as resolved in the deconvolution spectra and discussed in detailed. When come to the O1s spectra it is very intensive than the C1s. The intensity ratios of the C1s and O1s peaks are 1:3 and the C/O has calculated to be 1.84. It is due to the more amounts of oxygen functional groups present in the GOQDs and the great success of the oxidation of the BC in the presence of green oxidizing agent H2O2 under microwave. Such effectiveness in oxidation within a 2 min is great advantage of this method of GOQDs fabrication. The extent of oxygen functional groups present in GOQDs is highly helpful for water solubility and bioimaging applications. The deconvolution results from the literature are explained that the single broad O1s peak further displayed two distinct peaks at 531.6 and 533.1 eV corresponding to the C=O and C-OH oxygen functional groups on the surface of the GOQDs derived from black carbon [Green chem. 2017, 19, 900-904, Anal Chem. 2017, 89, 12520-12526 and Qin et al. Nanoscale Res Lett. 2021, 16, 14]. The peak at 531.6 nm has less intensity than 533.1 eV, is an indication that the GO surface has less enriched with many carbonyl functional groups than alcohol and epoxide kind of oxygen functionalities. Please check line no 227. The same results we obtained in our deconvoluted O1s spectrum. Please check line no 242. The spectrum is here for your reference.

11.Why do the authors conclude that “The reported GOQDs synthesized by the green synthesis method has excellent blue emission at 400 nm with 25 nm size, which is greatly suitable for many biological applications such as in vitro and in vivo imaging and therapy”?

Response:

Thank you for your critical question. We have claimed our results that the GOQDs are suitable for in vitro and in vivo imaging with a size of 20 nm. The text was modified accordingly as “The reported GOQDs synthesized by the green synthesis method have blue emission at 400 nm with 20 nm size, which is suitable for in vitro and in vivo imaging”. Please refer to line no 265. Previously we have mentioned the therapy along with imaging as we are aiming to use the GOQDs to cure colon cancer by functionalizing the GOQDs with suitable drugs. To meet this end, we also revise the manuscript title to “Green synthesis of blue-emitting graphene oxide quantum dots for in vitro CT26 and in vivo zebrafish nano-imaging as diagnostic probes” Please refer to line no 1.

  1. Are the dosages of GOQDs chosen for cytotoxicity experiments suitable? What is the basis for the dosage choice? In a real application, what dosage of GOQDs will take to achieve the best performance?

Response:

Thank you for your critical question. IC50 is the measured choice for assessing the toxicity of nanomaterials. According to this at what concentrations of nano materials, 50% of the cells are viable is the indicator of the toxicity. Usually most of the toxicity experiments use µg/ml of nanomaterials to inject to the cells or mice [RSC Adv., 2016, 6, 89867-89878]. However, depends on the level of toxicity, the values may lower (ng/ml) or higher (mg/ml) to get the IC50 values to the particular nanomaterials and cells [Scientific Reports, 2017, 7, 12896, Scientific Reports, 2019, 9, 4101, Biophys Rev. 2020 ,12, 703–718]. In the most adopted cases µg/ml has been reported in various cell lines and in mice [RSC Adv., 2016, 6, 89867-89878 and Biomaterials 2014, 35, 5041-5048]. Compared to the literature we found that the IC50 of GOQDs prepared from BC by microwave method are very high. We even observed that ~90% cell viability at 200 µg/ml. Hence the GOQDs reported in the current research are highly biocompatible. HeLa cells (cervical cancer cells) and A549 cells (lung carcinoma cells) have received the greatest attention for toxicity studies; other cells, including MCF-7 (breast cancer cells), red blood cells, and stem cells, have also been investigated. Mice, zebrafish, Caenorhabditis elegans, and green gramme sprouts have all been used in in vivo experiments for the study. In the literature, in vitro testing is more prevalent than in vivo tests overall [RSC Adv., 2016, 6, 89867-89878]. Based on the aforementioned results, the dosages of GOQDs chosen (0, 25, 50, 100, 200, and 400 μg/mL) for cytotoxicity experiments suitable is suitable. Please refer to line no 161.

  1. Will the zebrafish emit fluorescence under UV radiation? Will the background fluorescence disturb the experiments?

Response:

Thank you for your critical question. Usually, some models of zebrafish are blue light emitters under UV light. However, Danio rerio is not. Though it may have a blue background, that is not strong enough to dominate the luminesce from the GOQDs. It was very appealing in the in vitro studies that the GOQDs are very strong blue light emitters in CT26 cells upon excitation. Anticipating that the same will occur in the mice models in our future studies. Hence the background does not disturb the experiments.

A recent study by Liu et al. reported that blue-emitting CDs are good fluorescent markers in zebrafish. The bioimaging of zebrafish larvae under bright field and fluorescence revealed that the blue fluorescence from the CDs they fabricated. It was further evidenced by the merged images of bright and fluorescence images of Danio rerio larvae confirming that the blue emission was purely from the CDs not from the background or zebrafish. [ACS Appl. Mater. Interfaces 2020, 12(43), 49012–49020.] Hence, we are very confident that the blue emission from GOQDs is inside the cells and zebrafish as well.

  1. What is the relation between CT26 and zebrafish?

Response:

Thank you for your informative question. We choose the CT26 colon cancer cells as existing cell models in our institution. Further, we would like to use the GOQDs for cancer theranostics against colon cancer. i.e. as a preliminary imaging probe, we tried against colon cancer cells. As a future study, we would like to use it as a therapeutic probe by functionalizing chemotherapeutic and photosensitizers on GOQDs to eradicate colon cancer by adopting multiple therapeutic methods. We have chosen zebrafish as an in vivo model to demonstrate the imaging capability of GOQDs, further, we would like to extend the experiments to the mice models. In another sense, we would like to use GOQDs for water purification studies to remove heavy metals and organic pollutants. As an aquatic model, we have chosen Danio rerio to study the toxicity of GOQDs [ACS Sustainable Chem. Eng. 2013, 1(5), 462–472].

Danio rerio is a valuable non-mammalian vertebrate model that is frequently used to investigate disease, including more recently cancer. The significant evolutionary conservation of cancer-related genes between zebrafish and humans enables the extrapolation of study findings from fish to people. Zebrafish have drawn interest as a reliable model for cancer research due to their high fecundity, economical care, ability to dynamically visualize tumor progression in vivo, and ability to conduct chemical screening in large numbers of animals at affordable expenses. To understand the pathophysiology of a complicated disease like cancer, it may be crucial to look at both genetic and epigenetic changes. Zebrafish has proven to be a trustworthy model to research human cancer and may be appropriate for assessing the invasiveness of patient-derived xenograft cell lines, according to recent advancements in zebrafish xenotransplantation studies and medication screening. Rapid, extensive analysis of zebrafish in vivo drug reactions and kinetics will likely result in novel applications in combination therapy and personalized medicine. Zebrafish are on the verge of joining the mouse as a pre-clinical cancer model due to all of the aforementioned factors. Nevertheless, due mostly to the largely conserved mammalian genome and biological processes, the mouse will continue to be useful in the final stages of pre-clinical drug screening [Genes 2019, 10(11), 935]. We would like to investigate the elimination of colon cancer in zebrafish using modified GOQDs, which could serve as an imaging and therapeutic probe.

Reviewer 2 Report

Please see the attached PDF.

Author Response

Point-by-point response to Comments and Suggestions from Reviewer 2

Thank you for reviewing our work and positive comments. We have made the suggested changes in the revised manuscript which are highly important to improve the quality and readability of the manuscript.

In the article “Green synthesis of graphene oxide quantum dots for in vitro and in vivo biocompatible fluorescent imaging probes” by Gorle et al, the authors presented a green microwave-assisted synthesis of graphene oxide quantum dots (GOQD) using black carbon and H2O2 as green oxidizing agents. This green synthesis method is very fast (2 mins) and environmentally friendly as it doesn’t require any hazardous acidification process or any additional trace metallic removal process. The synthesis yielded 20-25 nm size GOQD that display strong characteristics and great biocompatibility as demonstrated in in vitro and in vivo studies. The topic and content of the article are interesting, especially because carbon-based quantum dot synthesis is an emerging field, and synthesizing nanomaterials in a green and biocompatible way is a great interest to many researchers. Due to these reasons, I recommend publishing this article in MDPI Pharmaceutics after addressing the following comments:

Response:

Thank you for complementing the article is interesting and recommending to publish in MDPI, Pharmaceutics. Yes, we agree that the carbon based QDs are emerging in nanotechnology and we have given an attempt to produce GOQDs in 2 min short time by adopting the green chemistry principles.

  1. There is a lack of innovation in the study. Kang et al. published a manuscript in 2019 (https://www.nature.com/articles/s41598-018-37479-6#Abs1) under the title of “Graphene Oxide Quantum Dots Derived from Coal for Bioimaging: Facile and Green Approach”. In the manuscript, they used a facile pulsed laser ablation in liquid (PLAL) technique for preparing GOQDs in 5 min using coal as a precursor and ethanol. The authors failed to cite this article in Table 1.

Response:

Thank you for your critical comment. The novelty of this manuscript is clearly stated that we advanced in the GOQDs preparation by shortening the time down to 2 min. reducing hazardous chemicals like strong concentrated acids and oxidizing agents like KMnO4. These chemicals release poisonous gases and cause metal impurities in the final product. The metal impurities may interfere when it is used in biomedical and sensor applications and deviate from the final results. It was clearly stated in the manuscript. Moreover, the starting material BC is very economic to prepare a large-scale synthesis of GOQDs. A microwave oven is more accessible and scalable than a pulsed laser ablation system We have cited the suggested research article (Ref 46) in our revised version.

  1. Author stated that by using H2O2 as an oxidizing agent, the contamination caused by metallic impurities can be prevented. But, H2O2 is also considered as a toxic chemical which easily breaks into free radicals. How are the H2O2 contents removed from the final product?

Response:

Thank you for your valuable question. According to our knowledge as well as from the literature, H2O2 is a green oxidizing agent [Hydrogen Peroxide as a Green Oxidant for the Selective Catalytic Oxidation of Benzylic and Heterocyclic Alcohols in Different Media:

An Overview]. By replacing the existing oxidizing agents with H2O2 can avoid the metal impurities and the OH radical generated in the reaction will interact with C=C bonds in the carbon and will convert into alcohols (C-OH), diols (OH-C-C-OH), ethers (C-O-C) and carboxylic acids (–COOH). As a result there is no possibility of free H2O2 residues. If there is any excessive of H2O2, it can be removed by simple washing.

  1. The reaction requires 50 mg of BC powder and 50 ml of H2O2 What is the product yield using green synthesis?

Response:

Thank you for your informative question. We have obtained 5 mg of the end product at the end of the purification.

  1. The author argues that the resultant GOQD yields uniform size distribution of 20-25nm. Size distribution data or low mag TEM images should be provided to strengthen this

Response:

Thank you for your constructive comment. We are unable to provide the low magnification TEM at this moment of revision. We have considered new image for Figure 2a to measure the size distribution in the revised manuscript and provided the size distribution curve. Here is the image for your reference (Figure 2a)

  1. Author stated that GOQD has very good water solubility. It is recommended to provide backup data (e.g., DLS) to strengthen this argument.

Response:

GO contains many oxygen functional groups. As a result it is water soluble and that is a well known fact from the literature. Nanomaterials with smaller sizes improve the surface charge. Hence display the water solubility without any or limited oxygen functional groups. Example graphene quantum dots form GO after reduction. Due to the small seizes of the GOQDs the size to charge ratios, undoubtedly the GOQDs are highly water dispersible/soluble. It was well quoted in most of the literature reports and reviews [Acc Chem Res 2013,15;46(10):2211-2224, Front. Chem., 2020, Sec. Nanoscience].

  1. Author claims that the GOQD has strong photostability, but there is no experiment designed for the photostability An in vitro experimental design testing for the photostability of the GOQD at various environments will be a good addition.

Response:

Thank you for your valuable suggestion. We have displayed the luminescence in water and in the in vitro experiments in PBS. More over the GOQDs are treated with PBST and Triton X- solution before recording the CLSM. Hence indirectly the photostability in different chemicals are tested as a preliminary concern. In addition, the in vivo zebrafish experiments reveal that the internalized GOQDs are disclosed its biological stability in a real time environment. However, we will also consider the respected reviewer valuable comments in our next communication.

  1. Scale bar in Figure 1B is

Response:

Thank you for your valuable suggestion. We have adjusted the scale bar in Figure 1.

Figure 1. Electron microscope images of BC before microwave irradiation (a) SEM. (b) TEM

  1. Scale bars and axes of Figure 2 are not

Response:

Thank you for your valuable suggestion. We have adjusted the scale bars in a better manner in the revised manuscript.

  1. The de-convoluted spectrum of C1s should Figure be labeled in Figure Need a legend.

Response:

Thank you for your valuable suggestion. We have added the suggested changes in the Figure 3c.

  1. Page 7, line 223: The authors mentioned “The peak at 200 nm corresponded to n→π* transition and is the indication of C=C and the peak at roughly 320 nm related to π→π* transitions of C=O in GOQDs. However, the peaks around 200 nm and 300 nm are due to π-π* transition and n- π* electronic transitions, This needs to be corrected. (https://onlinelibrary.wiley.com/doi/full/10.1002/smll.201601001?casa_token=iX93jbLZ7sYAAAAA%3AhLctOIkhcRniyNNN2_SmGWERkkh2e6CZXC0gYCr8L9qz3rusnegqexcKIn qpzIOf2vG0WGXo34Fvvew).

Response:

Thank you for your careful evaluation and valuable suggestion. We have resolved the suggested issue in the manuscript. Please check the lines no 245 to 247.

  1. Figure 4A, absorbance is spelled Additionally, proper legends are missing.

Response:

Thank you for your valuable suggestion. We have corrected the absorbance and provided the appropriate legend in the revised manuscript.

  1. It seems that GQD has a higher quantum yield than Synthesis process of GQD is also fast and simple (ACS Appl. Nano Mater. 2018, 1, 4, 1623–1630). What are the advantages of using GOQD over GQD?

Response:

Thank you for your critical question. The current method involves the usage of simple house hold microwave than reported. Both the GQDs and GOQDs have their own advantages as both are originated from same carbon source and functionality. However, the GOQDs have many surface oxygen functional groups. As a result, it has obvious water solubility and favors covalent and non-covalent bio-conjugation than GQDs. The quantum yields of GOQDs also good enough to be an imaging probe. It was evident from PL, in vitro and in vivo imaging. In addition, it is another simple method to generate GOQDs in an ecofriendly and economic way.

  1. Page 9, line 247: The equivalent of 0.05 mg/mL to 0.25 mg/mL is 50 to 250 ppm, respectively. The written range in the text does not match the ppm concentration in Figure 5A. This needs to be corrected.

Response:

Thank you for your careful evaluation and suggestion. We have made the necessary changes in the text as well as in the Figure 5A in the revised manuscript. Please check line no 270.

  1. Figure 5A: Statistical analysis must be

Response:

Thank you for your valuable suggestion. We have provided the statistical analysis in the Figure 5A in the revised manuscript.

  1. Figure 6 needs to be discussed in better It is not evident to visually see the differences between images to provide the insight that the authors aimed to claim.

Response:

Thank you for your constructive comment. We are very clear with our discussion and the results presented. We claimed that the GOQDs are biocompatible due to no any visible changes in the morphologies of the zebrafish at any stage or even well grown zebrafish after 96 hpf. It was true for 0-400 µg/ml of the GOQDs. The morphology of GOQDs treated zebrafish is same as the control / untreatedzebra fish.

Please check lines no 323 to 334.

  1. Nanoparticles with size 20-25 nm are more likely to be up taken by cell through endocytosis pathway (Chem. Soc. Rev., 2021, 50, 5397–5434). More explanation on GOQD uptake through phagocytosis mechanism is recommended.

Response:

Thank you for your careful evaluation and valuable suggestion. We have corrected the given suggestion and agree with the reviewer. Please check lines no

… These images show that the synthesized GOQDs are internalized through a non-targeted endocytosis internalization mechanism and are well distributed throughout the cytoplasm and nucleolus [76]. Please check the lines no 278 to 280.

  1. Author explained about advantages of NCD and CQD in line 319-335 and concluded that GOQD can contribute in a same way. However, the listed factors (e.g., passing through BBB, biodistribution ) are affected by so many other different variables. More detailed explanation is required to link the relation between NCD/CQD with GOQD.

Response:

Thank you for your question. Here we presumed that the GOQDs may act like NCD and CQDs as there are many similarities in physio-chemical properties in the future. The authors would like to exploit their results in other fields by citing relevant literature. It is completely a comparison only it is not at all a conclusion without any clear evidence. We would like to investigate the elimination of colon cancer in zebrafish using modified GOQDs, which could serve as an imaging and therapeutic probe in our future studies. Please check lines no 373 to 376.

  1. Proofreading the draft is strongly suggested. For examples, please check typos in lines 253, 261, and 331.

Response:

Thank you for your valuable suggestions. We have given the careful proofread throughout the manuscript.

Reviewer 3 Report

The authors provide an environment-friendly method for the preparation of GOQDs and their application to fluorescence imaging of cells and also zebrafish. The authors not only provide characterisation data for the products obtained by this method of preparation, but also compare their properties with those of other carbon nanomaterials. There are some formatting errors and unexplained parts in the text, which are recommended to be corrected for acceptance.

1. The dose ranges and units used in the two biological models, cellular and zebrafish, are not consistent.  The authors need to provide an explanation for this and revise it.

2. Why were CT26 tumor cells selected for testing the safety of GOCDs?

3. The fluorescent images in the article alone are currently not sufficient to demonstrate the mechanism by which GOCDs enter the cells and more experimental results are needed.

4. The formatting errors in the article are: in vivo is not italicized, and the scale bar in fig1 and fig7 is not clearly marked. It is suggested that the images of zebrafish in fig 7 should be resized to a uniform size.

Author Response

Point-by-point response to Comments and Suggestions from Reviewer 3

Thank you for reviewing our work and positive comments. We have made the suggested changes in the revised manuscript which are highly important to improve the quality and readability of the manuscript.

  1. The dose ranges and units used in the two biological models, cellular and zebrafish, are not consistent. The authors need to provide an explanation for this and revise it.

Response:

Thank you for your valuable suggestion. We apologize for the mistake happened. We mentioned the dosage of GOQDs in ppm for in vitro studies and mg/ml for in vivo zebrafish. The revised version has resolved the issue and mentioned the concentrations in mg/ml in all the relevant places.

Here are images for your reference

Figure 5. (a) Cell viability assay of GOQDs at different concentrations. Confocal fluorescence images of GOQDs in CT26 cells. (b) Bright-field image. (c) Blue emission of GOQDs within the cells recorded at 405 nm excitation. (d) The overlay of bright-field (b) and fluorescence (c) images. (The concentration of GOQDs used was 200 µg/ml).

Figure 6. Optical microscopy images of representative developmental stages of zebrafish embryos in water containing different concentrations of GOQDs at different times.

  1. Why were CT26 tumor cells selected for testing the safety of GOCDs?

Response:

Thank you for your informative question. We have chosen CT26 colon cancer cells for the fluorescence diagnosis purpose. Not just to evaluate the toxicity alone. We intend to use the GOQDs for therapeutic applications in future studies by functionalizing with suitable chemotherapeutic molecules along with GOQDs-mediated gene therapies. As a prior concern, we have tested the compatibility of GOQDs in CT26 cells.

  1. The fluorescent images in the article alone are currently not sufficient to demonstrate the mechanism by which GOCDs enter the cells and more experimental results are needed.

Response:

Thank you for your valuable suggestion. Yes, we do agree with your curiosity. However, as a preliminary demonstration, we have seeded the CT26 cells with various concentrations of GOQDs. As the GOQDs do not functionalize with any targeting agent, we anticipated that the GOQDs enter into the cells by simple endocytosis pathway, and it was reported the same in literature [Small 2019, 1902136 and Chem. Soc. Rev., 2021, 50, 5397–5434]. However, we will consider your suggestions in our future studies and will provide insights into the GOQDs internalization mechanism in a detailed study.  

  1. The formatting errors in the article are: in vivo is not italicized, and the scale bar in fig1 and fig7 is not clearly marked. It is suggested that the images of zebrafish in fig 7 should be resized to a uniform size.

Response:

Thank you for your valuable suggestions. We have italicized the in vitro and in vivo and where ever necessary in the revised manuscript. We have clearly marked the scale bars in all the figures and resized the suggested zebrafish image for your reference.

Figure 1. Electron microscope images of BC before microwave irradiation (a) SEM. (b) TEM.

Figure 7. Distribution of GOQDs inside a fully developed (96 hpf) zebrafish. (a) optical image of control (0 mg/ml), (b) fluorescence image of treated (100 mg/ml).
